# Mid-Infrared Response from Cr/n-Si Schottky Junction with an Ultra-Thin Cr Metal

**DOI:** 10.3390/nano12101750

**Published:** 2022-05-20

**Authors:** Zih-Chun Su, Yu-Hao Li, Ching-Fuh Lin

**Affiliations:** 1Graduate Institute of Photonics and Optoelectronics, The Department of Electrical Engineering, National Taiwan University, Taipei 10617, Taiwan; f06941006@ntu.edu.tw (Z.-C.S.); r09941015@ntu.edu.tw (Y.-H.L.); 2Graduate Institute of Electronics Engineering, The Department of Electrical Engineering, National Taiwan University, Taipei 10617, Taiwan; 3Department of Electrical Engineering, National Taiwan University, Taipei 10617, Taiwan

**Keywords:** nanomaterial, metal–semiconductor Schottky interface, silicon-based photodetector, mid-infrared, scanning electron microscope, atomic force microscope, X-ray diffraction, rapid thermal annealing, hot carrier effect, surface plasmon polariton, localized surface plasmon

## Abstract

Infrared detection technology has been widely applied in many areas. Unlike internal photoemission and the photoelectric mechanism, which are limited by the interface barrier height and material bandgap, the research of the hot carrier effect from nanometer thickness of metal could surpass the capability of silicon-based Schottky devices to detect mid-infrared and even far-infrared. In this work, we investigate the effects of physical characteristics of Cr nanometal surfaces and metal/silicon interfaces on hot carrier optical detection. Based on the results of scanning electron microscopy, atomic force microscopy, and X-ray diffraction analysis, the hot carrier effect and the variation of optical response intensity are found to depend highly on the physical properties of metal surfaces, such as surface coverage, metal thickness, and internal stress. Since the contact layer formed by Cr and Si is the main role of infrared light detection in the experiment, the higher the metal coverage, the higher the optical response. Additionally, a thicker metal surface makes the hot carriers take a longer time to convert into current signals after generation, leading to signal degradation due to the short lifetime of the hot carriers. Furthermore, the film with the best hot carrier effect induced in the Cr/Si structure is able to detect an infrared signal up to 4.2 μm. Additionally, it has a 229 times improvement in the signal-to-noise ratio (SNR) for a single band compared with ones with less favorable conditions.

## 1. Introduction

The mid-infrared (MIR) spectrum, covering wavelength band from 2 to 25 µm, is of interest to many scientific investigations. In addition to being an important main absorption band for atmospheric molecules (3–5 µm and 8–12 µm) and human body temperature sensing, MIR is used as a fingerprint band for many chemicals [1,2,3,4]. Therefore, it is very attractive to develop MIR light-detection technology. MIR light sensing is an important technology that may be involved in applications such as thermal image sensing [5], hazardous gas detection [6,7], biomedical sensing [8], microphysics [9], self-driving car technology, and so on. There have been many choices of materials for mid-infrared detectors. To achieve the detection of longer wavelength signals, narrow energy gap semiconductors (PbSe, 0.165 eV [10], PbTe, 0.190 eV [11], InGaAs, 0.354 eV [12], InSb, 0.17 eV [13]), two-dimensional (2D) graphene [14,15], and novel black phosphor materials [16,17] became the major targets in research to detect lower energy photons. However, narrow-gap semiconductors require epitaxial growth and are very expensive for use in daily life; large-area 2D graphene materials require consideration of process quality; and black phosphor materials require heterogeneous structures with other materials for better response to infrared light. In addition, all of these materials are extremely difficult to integrate with silicon-based components. In recent years, breakthroughs in nanophotonic technology have led to greater developments in the field of optoelectronics [18,19]. Through advanced microelectronic process technology [20], silicon-group nanophotonic technology is expected to perform multifunctional components in one single chip. Such a low-cost multiprocessing technology has considerable market advantages.

As a common semiconductor component, Si-based Schottky devices are widely used in the fields of logic analysis and integrated optoelectronics owing to their fast response rate and low background noise. Considering the excellent optical and electrical properties of thin-film Schottky components, increasing research has shown interest in developing infrared Schottky detectors [21,22,23,24,25,26,27,28]; for example, Yu et al. discussed the responsivity of copper/silicon/platinum Schottky optical detectors in the communication band and achieved a 0.542 mA/W response in the communication band by optimizing the Schottky barrier height and series resistance [21]. Hong et al. investigated the effects of light incidence from the metal or semiconductor side as well as temperature on the silver thin-film Schottky interface to detect infrared light. After increasing the Schottky interface absorption of the 1550 nm band light source, the response was enhanced accordingly, from 0.021 mA/W to 1.242 mA/W [28]. These results already break the detection cutoff wavelength (1.1 μm) of the silicon material. In recent years, among the many emerging theories, the study of hot carrier effect induced by surface plasmon (SP) has attracted the most interest. SP is a phenomenon of collective oscillation of electrons on metal surfaces due to external electromagnetic fields. It has different photoelectric properties from those of bulk materials [29,30,31,32]. For example, when SPs are coupled to a photon, an extremely strong SP will be generated. Additionally, the interactions between the plasmon and the photon might lead to subsequent energy transfer [33,34,35,36,37,38]. The carriers in the metallic thin-film layer are converted into hot carriers after absorbing longer wavelength photons. Once the hot carrier energy is larger than the Schottky barrier, the current can be formed for observation [39,40,41,42,43,44]. Based on the hot carrier theory, it would be beneficial to clarify the process of photoelectric conversion at the Schottky interface by investigating the association between the optical response and the physical properties of the nanometal surface.

Many studies have been carried out to fit the voltage and current characteristic curves of Schottky devices by utilizing the thermionic emission model [45,46,47,48], and we are aware that there are many important physical parameters at the Schottky interface, including Schottky barrier height, series resistance, and ideality factor [46]. Intuitively, we perceive that “when the energy barrier is larger, the ability to detect infrared light is lower” and “when the series resistance is larger, the measured signal is smaller”. However, to enhance the response, we ought to investigate what causes the change in energy barrier and what makes the resistance value larger. From a macroscopic point of view, we only consider the metal work function, the doping of different silicon types, and the electron affinity to investigate the contribution of Schottky barrier to the infrared detection. However, from a microscopic point of view, the uniformity of the contact surface, the plasmonic resonance of the metal particles, and the different crystal orientation of the metal all have an impact on the optical response. In this work, the Cr/n-Si Schottky device was originally able to detect only the 3 µm infrared band with 49 SNR and 1.22 nA response current. After improving the surface uniformity, contact area, and metal orientation, we have not only improved the SNR at 3 µm by 229 times but also found that the device was able to detect the 4.2 µm signal with an SNR of 10.7, which is a 1.2 µm broadening of the detection spectrum compared with the device without annealing.

## 2. Materials and Methods

In the fabrication process, n-doped silicon was used as the substrate with a thickness of about 600 µm and a resistance value of 2–7 Ω/cm. The substrate was cut into a size of 2.5 cm × 2.5 cm and then cleaned with an acetone, methanol, and isopropyl alcohol solution for 5 min to remove the organic materials on the surface. Then, the substrate was cleaned with a BOE etching solution to remove the oxide and particles on the surface of the substrate. The cleaned substrate was placed in the electron beam evaporation equipment (ULVAC, Inc., Kanagawa, Japan), and the evaporation rate was adjusted to 0.1 Å/s to form a 10 nm-thick chromium metal film on the substrate. Next, 100 nm finger electrodes and back electrodes were vaporized to complete the device fabrication. Figure 1a shows the structure of the device. The finger electrode allows the incident light to irradiate at the Schottky interface while collecting the carriers generated by the active metal layer. Figure 1b illustrates the energy band diagram of the interface. The carriers at the metal side absorb light energy and form a carrier flow through internal photoemissions or tunneling as well as hot carrier effects.

In this paper, we aim to discuss how the surface homogeneity, contact area, and metal crystallographic orientation specifically affect the optical response in a Cr-Si interface. Thus, we performed rapid thermal annealing processes on Schottky interfaces at 500 °C, 600 °C, and 700 °C in 300 s. Under suitable temperature and time conditions, the annealing process can release the stresses accumulated in the metal layer during vapor deposition and improve the contact at the interface. However, when the process temperature is too high or too long, it may have adverse effects. In the next section, we introduce a response current measurement system to observe the infrared detection capability of the device under each condition. At the same time, the films under different annealing conditions were analyzed by scanning electron microscope (SEM), atomic force microscopy (AFM), and X-ray diffractometer (XRD) systems. At the end of the article, we also describe how the homogeneity of the metal particles, the enhancement of the SP resonance, and the lattice orientation affect the photoelectric conversion process.

## 3. Results

### 3.1. Response Current Measurement

The monochromator can select narrow band wavelengths from a broadband light source and the chopper controls the illumination and non-illumination measurement conditions of the Schottky device (Figure 2a). The spectra (Figure 2b) show the spectroscopic results for the 2, 3, and 4 µm monochromator settings. The current of device under dark conditions is termed the dark current, and the current of an illuminated device is the light current. The response current is the difference between the light and dark currents. Figure 2c–e show the current variation over time of each device illuminated by 2, 3, and 4 µm light. All of the devices with different parameters were able to detect 2–3 µm signals, but the signal amplitudes were different under the same measurement conditions. The device annealed at 500 °C had the best signal performance, followed by the device without annealing, the device annealed at 600 °C, and the device annealed at 700 °C. Moreover, in Figure 2f, we carefully measured the detection limit of the device after annealing at 500 °C. We found that the spectral detection capability of the device after annealing at 500 °C had an improvement of 1.1–1.2 µm compared with the device without annealing.

In Figure 3, we analyzed the data of each parameter device irradiated with and without an infrared light source. First, the response current was calculated by the current level with and without illumination. Then, the root mean square value of the oscillation of each signal was taken to evaluate the noise level of the device. Finally, the SNR value was obtained by squaring the ratio of response current to noise. After optimizing the device by the annealing process, the SNR of the device was enhanced up to 54 times under the 2 µm light source and 229 times under the 3 µm light source compared with the device before annealing. This was not only because of the higher response after annealing, but also because of the significant reduction in background noise after annealing. The root mean square value of the noise is about 0.2 nA without annealing, but it is reduced to 0.05 nA after annealing, which means that the annealing process has a beneficial effect on the metal/semiconductor contact surface. In the following sections, material analyses were performed on devices with different annealing parameters to investigate the surface image, surface uniformity, and crystallization changes of the metal during the annealing process. Ultimately, the results of these material analyses are used to interpret the reaction trends.
(1)$$\mathrm{SNR}={\left(\frac{\mathrm{Response}\;\mathrm{current}\;\left(\mathrm{nA}\right)}{\mathrm{Noise}\;\left(\mathrm{nA}\right)}\right)}^{2}$$

### 3.2. Scanning Electron Microscope Images

In an SEM, the electron beam generated at the tip of the electron gun was accelerated by an electromagnetic field and controlled to hit the sample at different locations. The surface of the sample was imaged through the conversion of the current signal. With this technique, we can observe the surface metal image at the nanoscale and discuss the changes of the metal film before and after annealing. By adjusting the magnification to 50,000 times, the image of the film surface can be observed. Figure 4 shows the electron microscope image of the surface of a 10 nm chromium metal film. From (a) to (d) are the metal surface without annealing, after annealing at 500 °C for 5 min, after annealing at 600 °C for 5 min, and after annealing at 700 °C for 5 min, respectively. At the nanoscale, the surface of the device gradually changes from a smooth, flat surface to an island-like metal distribution as the annealing temperature rises. In the non-annealed and annealed at 500 °C cases, the surface showed no significant variations. In the annealed at 600 °C cases, the metal started to accumulate on the surface of the device. From SEM images of a device annealed at 700 °C, island structures of 10–100 nm size can be clearly observed. Because of the limitations of electron microscopy, images smaller than a 10 nm scale cannot be displayed clearly, and the roughness cannot be quantified. Therefore, in order to quantify the roughness of the surface, we used an atomic force microscope for the next step.

### 3.3. Atomic Force Microscopy Images

Since electron microscopy was not able to clearly demonstrate scales smaller than 10 nm and quantify roughness. To quantify the surface roughness, AFM was also used in this research. AFM had many advantages over scanning electron microscopy. First of all, unlike electron microscopy, which only provides 2D images, AFM provides three-dimensional (3D) surface images. At the same time, AFM does not require any additional treatment of the sample, such as coating with copper, carbon, or platinum. The atomic force microscope provides a more detailed image of the metal surface. Figure 5 shows the atomic force microscope images of the surface of a 10 nm chromium metal film. From (a) to (d) are the metal surface without annealing, after annealing at 500 °C for 5 min, after annealing at 600 °C for 5 min, and after annealing at 700 °C for 5 min, respectively. From the analysis of AFM, we can compare the maximum difference of film surface for each parameter. The maximum differences of the film surface without annealing, after annealing at 500 °C for 5 min, after annealing at 600 °C for 5 min, and after annealing at 700 °C for 5 min are 6.88 nm, 2.34 nm, 6.1 nm, and 53.5 nm, respectively. The trend of the maximum height difference corresponds precisely to the trend of the response current. The ideal annealing temperature is 500 °C, followed by no annealing, 600 °C, and 700 °C in order.

In the calculation of surface roughness, the arithmetic mean roughness (abbreviated as Ra) is a method to calculate the mean relative height of each point on a plane. The formula is as follows:(2)$$\mathrm{Ra}=\frac{1}{A}\overset{A}{\underset{0}{\iint }}\left|Z\left(x,y\right)\right|\mathrm{dxdy}$$

Ra was calculated by summing the difference Z between each point of the rough surface and the average height and dividing by the area A. The results of Ra calculation and the maximum height difference are recorded in Table 1. The arithmetic mean roughness of the film surfaces without annealing, after annealing at 500 °C for 5 min, after annealing at 600 °C for 5 min, and after annealing at 700 °C for 5 min are 4.468 nm, 0.341 nm, 0.998 nm, and 8.438 nm, respectively. From Table 1, after annealing at 500 °C, the device has the smallest height difference and the lowest roughness. The next parameter is the annealing temperature of 600 °C, whereas the annealing temperature of 700 °C shows the largest height difference and the roughest surface. So far, it can be concluded that setting the film at a high temperature of 500 °C would indeed improve the electrical properties of the film by the more uniform and flat film due to surface lattice rearrangement. However, if the temperature continues to rise to 700 °C, the lattice arrangement and flatness of the film would be disturbed, which then leads to the island-like distribution surface. To verify the credibility of this suggestion, we performed XRD analysis.

### 3.4. X-ray Diffraction Measurements

X-ray diffraction analysis is a nondestructive analysis technique. Based on the interference pattern obtained from X-rays incident on the material at different angles, the material crystallization direction, crystallization quality, and the amount of stress present in the material can be inferred from known material information. When the crystallization of the material is better, the power loss of the electrons flowing in the film is lower and the response is higher. Figure 6 shows the XRD measurements of the films without annealing, after annealing at 500 °C for 5 min, after annealing at 600 °C for 5 min, and after annealing at 700 °C for 5 min. According to the data, two angles of Cr crystallization information can be found. The signal at an angle of 32.9 degrees corresponds to Cr <104> crystallization, and the signal at an angle of 61.71 degrees corresponds to Cr <214> crystallization. The annealing process is effective for both crystallization directions. After annealing at 500 °C, the <104> crystalline quality is increased quite effectively. Furthermore, the XRD peak strength change was converted into the change of microstrain and residual stress for each annealing parameter by Equations (3) and (4).
(3)$$D=\frac{K\lambda }{\mathrm{rcos}\left(\theta \right)}$$
(4)$$\sigma_{x}=\frac{E}{\left(1-\nu \right)}\times \frac{d-d_{0}}{d_{0}}$$

Among them, K, γ, θ, λ, E and υ are crystallite-shape factors (0.94), the line broadening at half the maximum intensity (FWHM), Bragg angle, X-ray wavelength (0.154 nm), Young′s modulus, and the Poisson′s ratio value (0.28), respectively. We show the results of the calculations in Table 2 and Table 3. The changes of microstrain and residual stress at each crystal plane can be quantified. The stresses after annealing at 500 °C are +0.071 GPa at the <104> crystal plane and −0.37 GPa at the <214> crystal plane, which is the lowest among all parameters. This result also confirmed that the annealing process at 500 °C indeed results in the best film quality.

## 4. Discussion

In review, two results can be summarized from the response measurement experiments. Firstly, the most uniform surface has the best optical response and the lowest noise when the thickness of metal is 10 nm. Secondly, in addition to the enhanced response, when the surface is more uniform, the energy photon detectable is lower. Figure 7 illustrates the relationship between these two phenomena and the resistance of the metal film, which displays the case of uniform surface (the film after annealing at 500 °C) and the case of island-distributed surface (the film after annealing at 700 °C). In the case of the island-distributed surface, the carriers generated after photoelectric conversion were not collected directly by the finger electrode. Electrons need to be transmitted through metal-to-metal tunneling or induction. Therefore, the equivalent resistance is much larger than that of a uniform surface resulting in a smaller response. That is why devices without annealing and annealing at 700 °C were unable to detect signals larger than 3 µm and had a lower measured response.

The above discussed the effect of the surface film structure on the current. Next, we discuss the effect of the surface film state on the photoelectric conversion process. From the hot carrier theory [33,34,35,36,37,38], we can categorize the photoelectric conversion process into three steps: “the generation of SP by irradiating light on the metal surface”; “the generation of hot electrons and hot holes during the decay of the surface plasma”; and “the decay of the generated hot carriers with time”; all are shown in Figure 8.

Q_SPP_ and Q_LSP_ are the quality factors of surface plasmon polaritons (SPPs) and localized surface plasmon (LSP), respectively. The larger the Q value, the stronger the SP resonance intensity. The surface plasma resonance quality factors Q_SPP_ and Q_LSP_ of Cr, which are obtained by converting the refractive index (n) and extinction coefficient (k), are shown in Figure 9a [49,50]. Above 1 µm, the Q_SPP_ rises from 100 to 3400 and the Q_LSP_ also starts to have a value higher than 1. The larger values of the two quality factors for the mid-infrared band represent a considerable enhancement of the SP intensity in the mid-infrared and even far-infrared bands.

After determining the efficiency of SP resonance, the next step is to discuss the generation of hot carriers. The decay of SP in thin metallic films can be solved using the Fermi golden rule calculation [51]. In the Fermi golden rule calculation, a metal film of thickness *Lz* is assumed to be centered in the plane of z = 0 in Cartesian coordinates. By solving Maxwell′s Equations and integrating their field strengths, the transition rate of hot carriers in the symmetric and antisymmetric modes can be obtained. In this paper, the relationship between the film thickness and the hot carrier transition rate is simplified, as shown in Equation (5).
(5)$$\Gamma \propto \frac{1}{\frac{\left[\left(\gamma_{in}^{2}+k^{2}\right)\sinh(\gamma_{in}L_{z}\right)+\left(\gamma_{in}^{2}-k^{2}\right)(\gamma_{in}L_{z})]+[\sinh(\gamma_{in}L_{z})-\gamma_{in}L_{z})]}{2\gamma_{in}^{3}\left(\cosh\left(\gamma_{in}L_{z}\right)+1\right)}+\frac{k^{2}}{2\gamma_{out}^{3}}}$$

In Figure 9b, the curve of the hot carrier transition rate versus thickness can be calculated. Under the condition of a 10-times change in metal thickness, the hot carrier transition rate decreases with the increase in metal film thickness from 0.2 to 0.026, reducing by a ratio of about 90%. It has a significant impact on Schottky devices that generate current through hot carriers. In addition, the Schottky detector detects the infrared light by utilizing the interface formed by the metal–semiconductor (M-S) junction. However, when the light is exposed directly to the silicon substrate, the infrared light detection is not possible only through the energy gap of the silicon. The effective area is marked with yellow lines. The area surrounded by the yellow line is the area covered by the metal film, and the rest is the area without coverage, as shown in Figure 9c. According to the analysis of the software, the pixel ratio of the metal-covered area to the whole area is 24,223: 48,460. The area covered by the metal is about 50%, whereas the remaining interface lets the infrared light transmit directly into the silicon substrate. The response current is expected to be reduced by 50%.

The last issue to be considered is the carrier lifetime. Hot carriers are energetic carriers that in general tend to release energy and drop to the ground state. In this process, the loss of hot carriers mainly results from hot electron–phonon interaction and hot electron–cold electron interaction. In the hot electron–phonon interaction, heat is released due to lattice vibrations. In addition, the valence electrons in the metal, which are not bound to a specific location, form an electron sea. Because there are a large number of cold carriers (carriers with energy distribution at room temperature) in the electron sea, the hot carriers in metals decay instantaneously when they are generated. This is attributed to the energy exchange between the hot carriers and the large number of cold carriers in the electron sea. Many papers have widely used the two-temperature model to investigate the lifetime of hot carriers [52,53], and the lifetime of Cr is about 0.4–0.6 ps [54,55]. Figure 9d shows the result of the variation of the number of hot carriers with time. The number of generated hot carriers is decreasing logarithmically with time. The difference between the homogeneous film surface and the island distribution film surface shows at least 90% difference in the ratio of hot carriers.

In brief, we can explain that the reason for the better response of homogeneous films is not only due to the quality of the metal film–silicon interface but requires a step-by-step consideration of the physical parameters in the photoelectric conversion. In a uniform film, several nA values of response current can be detected, and the device has a background noise of 0.05 nA. After considering four physical parameters—surface resistance, film coverage, hot carrier generation rate, and hot carrier decay—we can estimate the reduction in the response current by 50%, 90%, and 90% for surface coverage, hot carrier transition rate, and hot carrier decay, respectively. The response current becomes 0.005 times smaller than the original response current. These smaller response currents are likely to be hidden by noise of 0.05 nA and unobservable in the experiment. That is why a metal film with a uniform surface has a better response and a longer cutoff wavelength.

## 5. Conclusions

In this research, the relationship between the optical response, the distribution, and the thickness of the surface metal was sorted out by material analysis data. Compared to island-distributed surfaces, both the signal and the cutoff wavelength of uniform nanometal thin-film devices are improved. The SEM and AFM image analysis showed that the metal coverage of the uniform surface and the island-distributed surface were 100% and 50%, and the roughness was 0.341 nm and 8.438 nm, respectively. These two characteristics influence the optical response of the Schottky devices significantly, resulting in at least 10-times change in response current. Under 2 µm irradiation, the response currents of the device were 0.2 and 13.7 nA for the island-like and uniform surfaces, respectively (with a difference of 68.5 times), while under 3 µm irradiation, the response currents of the device were 0.17 and 3.3 nA for the uniform and island-like surfaces, respectively (with a difference of 19 times). In addition to the improvement in response, the uniform surface film is capable to detect longer wavelength signals with a 10.7 signal-to-noise ratio for the 4.2 µm band. The reason for this result is twofold. First, since Schottky devices utilize the contact surface formed by the metal and the semiconductor to detect mid-infrared light, the less the metal coverage, the lower the response will be. Second, as the metal surface of the island distribution is thicker, it takes a longer time for the hot carrier to diffuse to the Schottky interface. Because of the hot carrier attenuation, the infrared signal becomes unmeasured by the device. In summary, we reveal the surface metal thin-film condition that determines the mid-infrared detection capability of silicon-based Schottky devices. Such a nanometal infrared detector is expected to be a candidate for mid-infrared detection technology owning to the convenient process method and suitable for integration with silicon-based CMOS devices.

## Figures and Tables

**Figure 1 nanomaterials-12-01750-f001:**
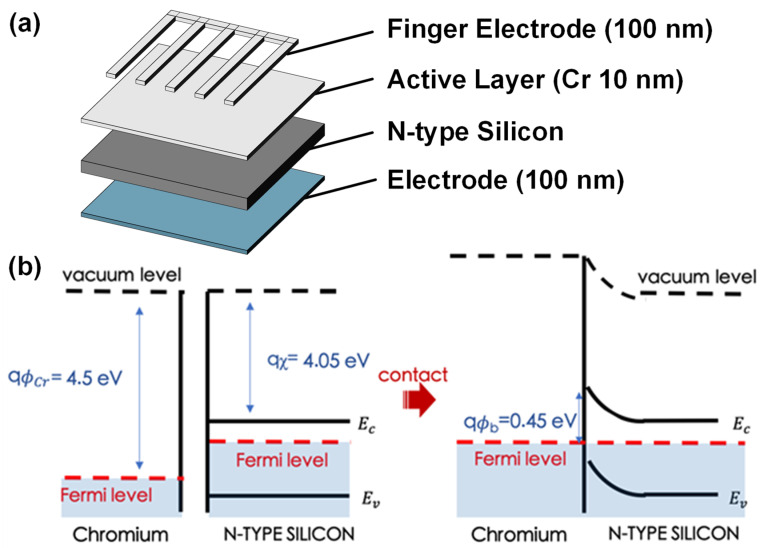
(**a**) The structure of the device; (**b**) the energy band diagram of the interface.

**Figure 2 nanomaterials-12-01750-f002:**
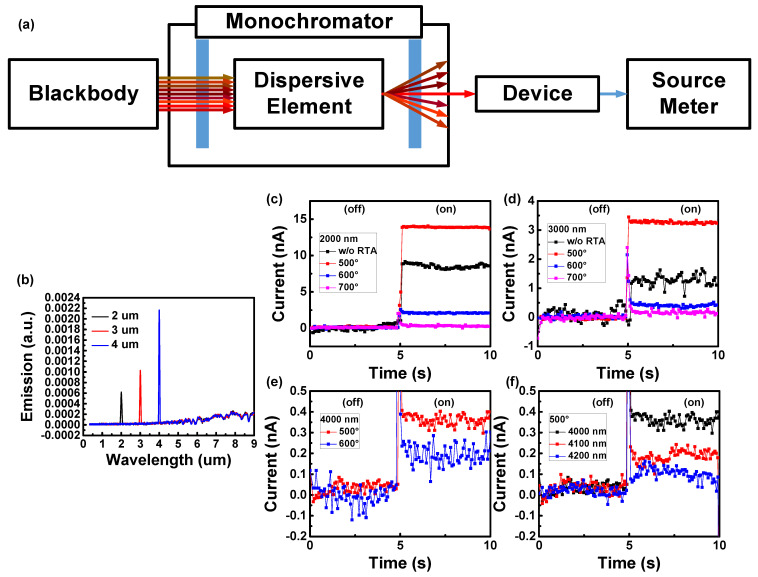
(**a**) Measurement setup with monochromator. (**b**) Spectroscopic results for the 2, 3, and 4 µm monochromator settings. (**c**–**f**) Current variation over time of each thermal annealing parameter device under 2 µm, 3 µm, and 4 µm. (**f**) After annealing at 500 °C, the behaviors of the device under 4, 4.1, and 4.2 µm light irradiation.

**Figure 3 nanomaterials-12-01750-f003:**
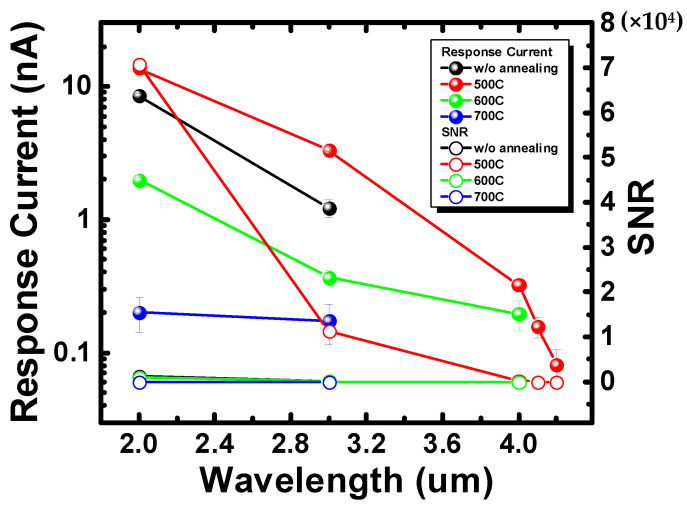
The response current and signal-to-noise ratio of devices with different annealing parameters.

**Figure 4 nanomaterials-12-01750-f004:**
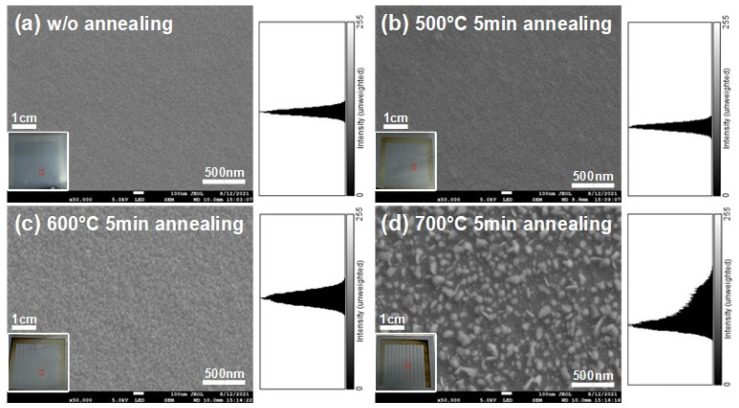
Scanning electron microscope images of the surfaces: (**a**) without annealing; (**b**) after annealing at 500 °C for 5 min; (**c**) after annealing at 600 °C for 5 min; (**d**) after annealing at 700 °C for 5 min.

**Figure 5 nanomaterials-12-01750-f005:**
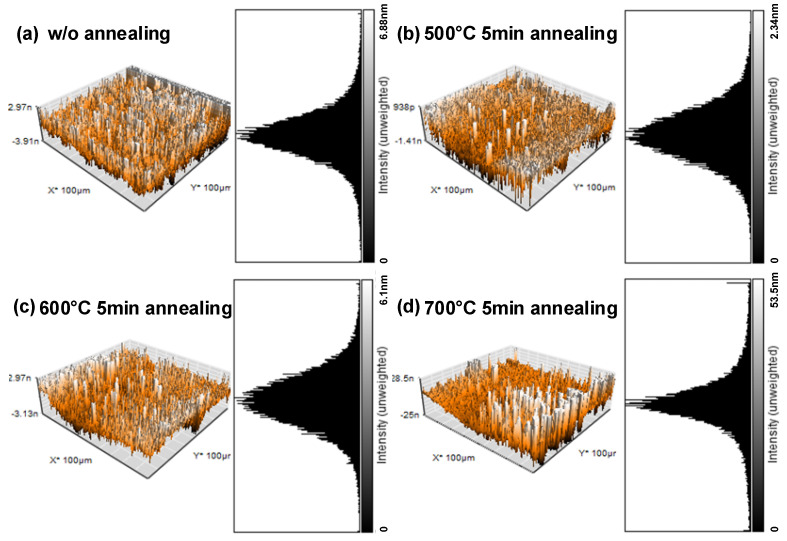
Atomic force microscope images of the surface: (**a**) without annealing; (**b**) after annealing at 500 °C for 5 min; (**c**) after annealing at 600 °C for 5 min; (**d**) after annealing at 700 °C for 5 min.

**Figure 6 nanomaterials-12-01750-f006:**
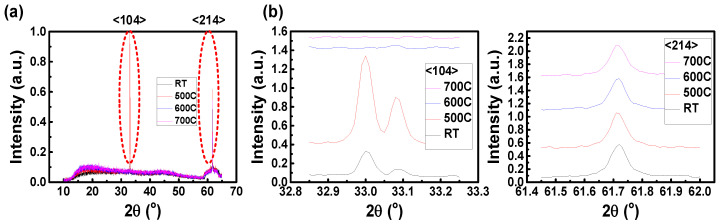
(**a**) The XRD measurements of the film without annealing, after annealing at 500 °C for 5 min, after annealing at 600 °C for 5 min, and after annealing at 700 °C for 5 min; (**b**) the XRD peak at the <104> and <214> crystal plane.

**Figure 7 nanomaterials-12-01750-f007:**
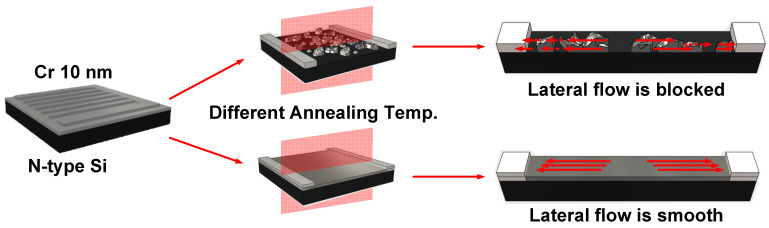
The uniform surface and the island-distributed surface.

**Figure 8 nanomaterials-12-01750-f008:**
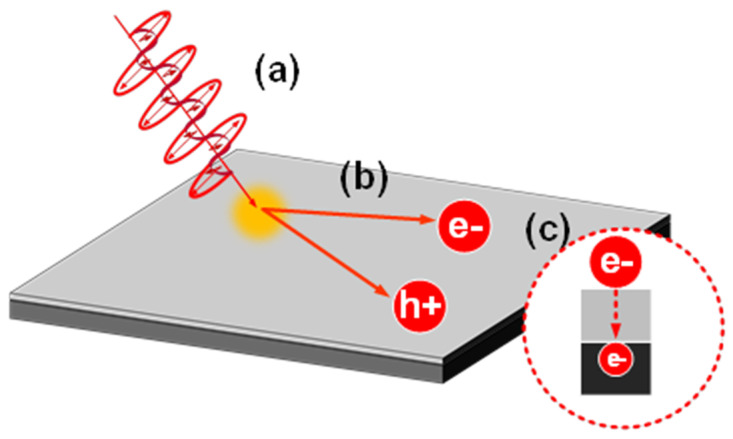
(**a**) The generation of surface plasmon by irradiating light on the metal surface. (**b**) The generation of hot electrons and hot holes during the decay of the surface plasma. (**c**) The decay of the generated hot carriers with time.

**Figure 9 nanomaterials-12-01750-f009:**
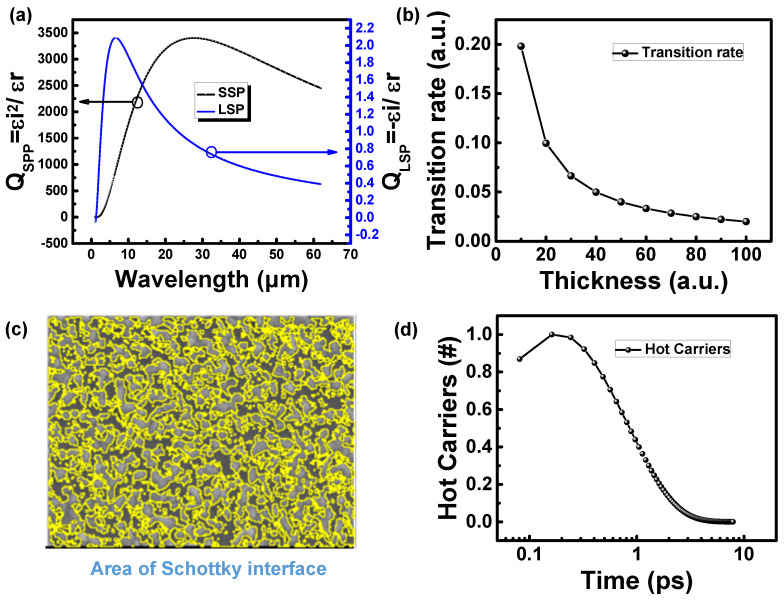
(**a**) Relationship between plasmon resonance quality factor and wavelength at Cr surface; (**b**) relationship between hot carrier transition rate and thickness; (**c**) schematic diagram of effective surface area; (**d**) decay of hot carrier number with time.

**Table 1 nanomaterials-12-01750-t001:** The results of atomic force microscopy.

Parameter	*w/o* Annealing	500 °C Annealing	600 °C Annealing	700 °C Annealing
Maximum different (nm)	6.88	2.34	6.1	53.5
Ra (nm)	4.468	0.341	0.998	8.438

**Table 2 nanomaterials-12-01750-t002:** The results of XRD at the <104> crystal plane.

Annealing Temperature (°C)	Crystallite Size, D_g_ (nm)	d-Spacing (Å)	Bragg Angle, 2θ (°)	Microstrain, ε	Residual Stress, σ_X_ (GPa)
R.T.	190.964	2.714	32.97	0.000185	0.077
500	201.559	2.712	32.99	0.000172	0.071

**Table 3 nanomaterials-12-01750-t003:** The results of XRD at the <214> crystal plane.

Annealing Temperature (°C)	Crystallite Size, D_g_ (nm)	d-Spacing (Å)	Bragg Angle, 2θ (°)	Microstrain, ε	Residual Stress, σ_X_ (GPa)
R.T.	104.047	1.5018	61.71	−0.0006	−0.25
500	73.2	1.5017	61.71	−0.0009	−0.37
600	95.789	1.502	61.7	−0.0006	−0.25
700	83.45	1.5018	61.71	−0.0007	−0.27

## Data Availability

The data presented in this study are available on request from the corresponding author. The data are not publicly available due to privacy concerns.

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
