# Peer review of "Mid-Infrared Response from Cr/n-Si Schottky Junction with an Ultra-Thin Cr Metal"

_nanomaterials, 2022, doi:10.3390/nano12101750_

Round 1

Reviewer 1 Report

In this study, the authors prepared Cr/n-Si Schottky junction for potential use as a mid-infrared sensor. In general, the article is interesting and can be considered for publication after a major revision. 

1) Provide an SEM image of the device part - i.e. Si, Cr, and finger electrodes should be visible. 
2) Chemical analysis should be provided by means of EDX elemental mapping (to confirm the uniform distribution of Cr on Si) and by XPS (to confirm that Cr was not partially converted to Cr oxide after high-temperature annealing). 
3) What was the measurement conditions for Figure 2 and 3? Is it room temperature and normal humidity? Please check the current response at different temperatures and humidity levels (i.e. devices must be stable in different parts of the world).
4) Have you tested the stability of the device, i.e. shelf-life? Typically, Cr layer and finger electrodes can be oxidized with time.   
5) I don't see any statistical data, i.e. how reproducible the results from each device and each measured point should be reported as a mean value + SD (add error bar charts). 
6) If possible, provide a time-resolved PL analysis - it will additionally help to understand the improvement mechanism. 

Reviewer 2 Report

I read the manuscript entitled “Mid-infrared response from Cr/n-Si Schottky junction with an ultra-thin Cr metal” by Zih-Chun Su et al, submitted for a publication in Nanomaterials.

In this work, the authors have investigated Cr-silicon-based Schottky device for infrared detection. In particular, the authors have reported the effect of surface/structure morphology, studied by SEM, AFM and XRD with different treatments, on the optical response of the Schottky device. On the basis of their study the authors have concluded that surface morphology has significant effect on the optical response of Schottky device.

The potential of Schottky devices in detection technology is well known and they have been object of several experimental and theoretical investigations for their basic characteristics and optical responses. It is certainly an important field of research to identify the physical parameters controlling the optical response of Schottky devices. Although similar studies have been reported in literature, the manuscript can still be a useful contribution in the specific field of research, however, the presentation of the experimental results needs improvement before judging its suitability for the publication. 

1) In the Fig. 4 the scales are not visible and the reader can have only a qualitative evolution of the morphology with the annealing conditions. I suggest the authors to provide corresponding histograms to let the readers able to see a quantitative evolution.

2) Similar problem is with the presentation of AFM images in Fig. 5. It is difficult to appreciate the morphological changes with the annealing. I think it is better to show two diminutional AFM images together with the z-axis line plots showing the roughness.

3) It is difficult to make any useful judgement on the basis of the XRD pattens shown in Fig. 6. I think the authors need to show the enlarged patterns and the line profiles. They can easily show the line profiles of the two reflections in such a way that the readers can make their own evaluations of the reported parameters.

4) Fig. 3 is almost information-less as the result is clear from the Fig. 2 itself. If the authors feel to show it graphically, it is enough to provide the signal and the noise in a single panel instead of using three different panels.

5) There are typos requiring corrections; for example SNR (Signal to Noise Ratio) should be defined before its use as acronym, Gpa => GPa etc.

Reviewer 3 Report

The paper reports a mid-infrared response from the Cr/n-Si Schottky junction with an ultrathin Cr metal.

The authors have discussed the topic in a very interesting way. The paper is well written and has a clear layout. The topics discussed in this paper should attract great interest from future readers. I must admit that I have not read such a well-prepared work for a long time.

I think the manuscript has sufficient scientific quality and relevance for Nanomaterials (ISSN 2079-4991). I suggest that you accept the publication as is.

Author Response

Thank you very much for the positive comments.

Round 2

Reviewer 1 Report

No more comments

Reviewer 2 Report

The manuscript is improved and I found the authors reply satisfactory. The revised manuscript is suitable for a publication.